# Novel Antioxidant Therapy with the Immediate Precursor to Glutathione, γ-Glutamylcysteine (GGC), Ameliorates LPS-Induced Cellular Stress in In Vitro 3D-Differentiated Airway Model from Primary Cystic Fibrosis Human Bronchial Cells

**DOI:** 10.3390/antiox9121204

**Published:** 2020-11-30

**Authors:** Chris K. Hewson, Alexander Capraro, Sharon L. Wong, Elvis Pandzic, Ling Zhong, Bentotage S. M. Fernando, Nikhil T. Awatade, Gene Hart-Smith, Renee M. Whan, Shane R. Thomas, Adam Jaffe, Wallace J. Bridge, Shafagh A. Waters

**Affiliations:** 1School of Women’s and Children’s Health, Faculty of Medicine, University of New South Wales, Sydney, NSW 2052, Australia; c.hewson@student.unsw.edu.au (C.K.H.); a.capraro@unsw.edu.au (A.C.); sharon.l.wong@unsw.edu.au (S.L.W.); n.awatade@unsw.edu.au (N.T.A.); a.jaffe@unsw.edu.au (A.J.); 2Molecular and Integrative Cystic Fibrosis (miCF) Research Centre, University of New South Wales and Sydney Children’s Hospital, Sydney, NSW 2052, Australia; 3School of Biotechnology and Biomolecular Sciences, Faculty of Science, University of New South Wales, Sydney, NSW 2052, Australia; gene.hart-smith@mq.edu.au (G.H.-S.); wj.bridge@unsw.edu.au (W.J.B.); 4Biomedical Imaging Facility, University of New South Wales, Sydney, NSW 2052, Australia; e.pandzic@unsw.edu.au (E.P.); r.whan@unsw.edu.au (R.M.W.); 5Bioanalytical Mass Spectrometry Facility, University of New South Wales, Sydney, NSW 2052, Australia; l.zhong@unsw.edu.au; 6School of Medical Sciences, Faculty of Medicine, University of New South Wales, Sydney, NSW 2052, Australia; b.fernando@student.unsw.edu.au (B.S.M.F.); shane.thomas@unsw.edu.au (S.R.T.); 7Department of Molecular Sciences, Faculty of Science and Engineering, Macquarie University, Sydney, NSW 2109, Australia; 8Department of Respiratory Medicine, Sydney Children’s Hospital, Sydney, NSW 2031, Australia

**Keywords:** cystic fibrosis, glutathione, redox, antioxidant, oxidative stress, *Pseudomonas aeruginosa*, LPS

## Abstract

Systemic glutathione deficiency, inflammation, and oxidative stress are hallmarks of cystic fibrosis (CF), an inherited disease that causes persistent lung infections and severe damage to the respiratory system and many of the body organs. Improvements to current antioxidant therapeutic strategies are needed. The dietary supplement, γ-glutamylcysteine (GGC), which is the immediate precursor to glutathione, rapidly boosts cellular glutathione levels following a single dose in healthy individuals. Efficacy of GGC against oxidative stress induced by *Pseudomonas aeruginosa*, which is a common and chronic pathogen infecting lungs of CF patients, remains unassessed. Primary mucocilliary differentiated airway (bronchial and/or nasal) epithelial cells were created from four individuals with CF. Airway oxidative stress and inflammation was induced by *P. aeruginosa* lipopolysaccharide (LPS). Parameters including global proteomics alterations, cell redox state (glutathione, oxidative stress), pro-inflammatory mediators (IL-8, IDO-1), and cellular health (membrane integrity, stress granule formation, cell metabolic viability) were assayed under six experimental conditions: (1) Mock, (2) LPS-challenged (3) therapeutic, (4) prophylactic (5) therapeutic and prophylactic and (6) GGC alone. Proteomic analysis identified perturbation of several pathways related to cellular respiration and stress responses upon LPS challenge. Most of these were resolved when cells were treated with GGC. While GGC did not resolve LPS-induced IL-8 and IDO-1 activity, it effectively attenuated LPS-induced oxidative stress and stress granule formation, while significantly increasing total intracellular glutathione levels, metabolic viability and improving epithelial cell barrier integrity. Both therapeutic and prophylactic treatments were successful. Together, these findings indicate that GGC has therapeutic potential for treatment and prevention of oxidative stress-related damage to airways in cystic fibrosis.

## 1. Introduction

The thiol antioxidant glutathione is produced and maintained by all cells of the body in the millimolar range of concentrations. Extracellular glutathione concentrations can vary greatly, with plasma levels typically in the low micromolar range. In the epithelial lining fluid (ELF) of the lung, total glutathione can be 100-times higher than plasma concentrations, with increases occurring in response to stimuli such as pathogens [1,2]. High glutathione concentrations in the ELF, and the ability to raise them, is likely an adaptive response to protect lung epithelial cells against cytotoxic damage caused by bacterial and viral infections [2]. Transport of glutathione out of lung cells is predominantly conducted by the apically expressed cystic fibrosis transmembrane regulator (CFTR) protein [3]. CFTR, which is a member of the ATP-binding cassette (ABC) family of membrane transport proteins, forms a chloride and bicarbonate channel. CFTR maintains homeostasis and fluid secretion for many tissues and organs including the lung, intestine, and kidney, along with sweat and pancreatic ducts.

Unsurprisingly, patients with cystic fibrosis (CF), which is an inherited disease caused by mutations in the CFTR gene, are characterised by a generalised systemic deficiency in extracellular glutathione and major lung function pathology [4]. CFTR dysfunction deregulates fluid secretion in the lungs, which leads to thick mucus formation and recurrent bacterial infections. Abnormally high levels of reactive oxygen species (ROS) with elevated pro-inflammatory markers are observed in the developing CF foetus even prior to exposure to microorganisms [5]. This innate oxidative state has been attributed to alterations in cellular proteostasis and the mitochondrial electron transport chain caused by deregulated processing and misfolded mutated CFTR. Release of toxins from bacteria that cause common and chronic respiratory infections such as *Pseudomonas aeruginosa* cell-surface lipopolysaccharide (LPS) further provokes an exaggerated neutrophilic inflammatory response that produces additional ROS [6]. The uncontrolled production of ROS can be detrimental through the promotion of aberrant cell signalling responses and the oxidative modification of biomolecules leading to lipid peroxidation, protein oxidation and dysfunction, and DNA strand-breaks, that together can promote loss of viability of lung epithelial cells [5].

Under normal physiological conditions, ROS generation and consequently oxidative stress is controlled by various adaptive cellular stress responses including the transient formation of cytoplasmic stress granules (SGs), activation of autophagy and alterations to mitochondrial function. ROS generation is balanced by several antioxidant defence systems, which include enzymes such as superoxide dismutase (SOD), catalase, peroxiredoxins (Prx), and glutathione peroxidases (GPx), which employ GSH as a substrate to detoxify ROS [5]. However, due to compromised glutathione transport into the extracellular environment, excessive ROS and inflammation are persistent in the CF lung.

In recognition of the systemic glutathione deficiency in CF, a considerable body of research has targeted glutathione supplementation as a therapeutic strategy to restore redox homeostasis in patients with CF [4,7,8,9,10]. Glutathione is a tri-peptide composed of glutamate, cysteine, and glycine that exists in a reduced (GSH) or oxidised (GSSG) state. It is synthesised in the cytosol of all cells by the activity of two enzymes: glutamate cysteine ligase (GCL) condenses glutamate and cysteine to form γ-glutamylcysteine (GGC) and glutathione synthase (GS) adds glycine to form glutathione. Homeostasis is regulated by feedback inhibition where glutathione interacts with GCL to modulate its activity. Glutathione exists in a reduced (GSH) and oxidised (GSSG) state.

It is thermodynamically unfeasible for exogenously administered glutathione to be passively taken up by cells due to the near 1000-fold higher total glutathione concentrations within the cell cytosol relative to that in the plasma [11]. Most cell types have a membrane-bound ectoenzyme, γ-glutamyltransferase, that effectively hydrolyses glutathione to its three component amino acids during cellular uptake. This means that administered glutathione and cysteine prodrugs such as *N*-acetylcysteine (NAC) simply provide substrates for GCL and thereby cannot increase cellular glutathione levels above homeostasis. An inability to increase above homeostasis, may explain why studies with oral and inhaled formulations of thiols (GSH and NAC) have repeatedly failed to demonstrate consistent improvements in CF clinical outcomes [7,8,12,13,14,15].

Alternatively, GGC can be taken up by cells intact where it is converted to glutathione [16,17,18]. Furthermore, GGC itself is an antioxidant by acting as a cofactor for GPx where it is effectively comparable to GSH for hydrogen peroxide detoxification [19] Animal safety trials have demonstrated that GGC is safe at a repeated limit dosage of 1000 mg/kg/day over a 90-day period [20]. Exogenous supply of GGC, which is sold as a dietary supplement in the USA, has been reported to suppress oxidative injury and improve mitochondrial function both in in vitro and in vivo animal models of oxidative stress-induced tissue damage [18,21,22,23,24]. In the context of lung disease, GGC has been tested in an LPS-induced mouse model of sepsis where it was demonstrated that GGC administration suppressed the production of LPS-induced inflammatory and oxidative mediators, which reduced lung tissue damage and sepsis lethality [24]. A randomised human pilot study confirmed that single doses of orally administered GGC could significantly increase GSH levels in lymphocytes of healthy individuals within 3 h, with a return to normal homeostatic levels by 5 h [16]. This oral bioavailability is in part attributed to the resistance of gamma amide linkage to protease mediated hydrolysis [25].

We hypothesise that administration of GGC as a therapy can modulate host oxidative and inflammatory responses elicited by the common CF pathogen *P. aeruginosa*. In this primary CF differentiated bronchial cell line study, we explored the potential of GGC as both a prophylactic or therapeutic treatment candidate for CF by host responses when treated with GGC before and after exposure to LPS.

## 2. Materials and Methods

### 2.1. Airway Epithelial Cell Culture

Human airway basal epithelial cells were obtained from brushing the nasal inferior turbinate and from bronchoalveolar lavage fluid (BALF) of four CF donors (2 male and 2 female; mean age 2.6 ± 0.9 years old) and one non-CF donor (male; 1.75 years old) during bronchoscopy. All CF subjects had homozygous DF508-CFTR genotype. The participants’ carers provided written informed consent. This study was approved by the Sydney Children’s Hospital Ethics Review Board (HREC/16/SCHN/120).

#### 2.1.1. Conditionally Reprogrammed Epithelial Cell (CREC) Culture

Primary airway cultures were established based on a previously published protocol [26]. At confluency, cells were dissociated using a differential trypsin method. Cells were either cryopreserved, or directly utilised for the 2D-conventional or the 3D-differentiated ALI cultures.

#### 2.1.2. 3D-Air-Liquid Interface (ALI) Differentiated Epithelial Cell Culture

CRECs were seeded at ~125,000 on Purecol^®^ Collagen I (Advanced Biomatrix, Carlsbad, CA, USA) coated 6.5 mm 0.4 µM Corning^®^ Transwell^®^ porous polyester membranes (Sigma-Aldrich, St. Louis, MO, USA). Cultures were supplemented with PneumaCult Ex-Plus Expansion media (Stemcell Technologies, Vancouver, BC, Canada) for 4–5 days. Once confluent, the media from the apical side was removed. PneumaCult ALI media (Stemcell Technologies, Vancouver, BC, Canada) was added on the basolateral side. Cultures were monitored for muco-ciliary differentiation over a period of 21–24 days before experiments were conducted. All experiments were carried out on cells of passage one or two.

#### 2.1.3. 2D-Conventional Epithelial Cell Culture

CRECs were seeded in a Purecol^®^ Collagen I coated (Advanced Biomatrix, Carlsbad, CA, USA) 96 well plate and supplemented with Bronchial Epithelial Growth Medium (BEGM) (Lonza, Basel, Switzerland). These cultures were used to assess stress granule formation and cell viability.

### 2.2. Sample Preparation

ALI differentiated bronchial CRECs from the CF donors (*n* = 4) were cultured at ALI and treated under six experimental conditions: (1) PBS control (Mock), (2) LPS-challenged (LPS+), (3) GGC added 24 h post-LPS challenge (therapeutic, T), (4) GGC added 24 h prior to LPS challenge (prophylactic, P), (5) GGC added 24 h prior and 24 h post LPS challenge (T+P); and (6) GGC alone. Cells were harvested in equal aliquots for mass spectrometry (proteomics) and oxidant-antioxidant content analyses. Experiments were performed in triplicate. The supernatant was saved for IL-8 cytokine and IDO-1 analysis. ALI differentiated bronchial and nasal cells were used for tight junction ZO-1 imaging. Stress granule and cell metabolic activity assays are compatible with 2D-conventional cultures requiring a large number of cells to initiate culture. With the limitation of primary bronchial epithelial proliferation capacity, assessment of stress granule and cell metabolic viability were carried out on the nasal epithelial cultures generated from the same four donors.

### 2.3. Mass Spectrometry

For the mass spectrometry analysis, total protein was determined by homogenising the cells in RIPA buffer (Life Technologies, Carlsbad, CA, USA) containing protease inhibitor cocktail (Sigma-Aldrich, St. Louis, MO, USA). Samples were sonicated using the Bioruptor Pico (Diagenode, Liège, Belgium) for a total of 10 min. Protein concentrations were determined using the 2-D Quant kit (Cytiva Life Sciences, Marlborough, MA, USA). Samples were reduced (5 mM DTT, 37C, 30 min), alkylated (10 mM IA, RT, 30 min) then incubated with trypsin at a protease:protein ratio of 1:20 (*w*/*w*) at 37 °C for 18 h, before being subjected to SCX clean-ups (Thermo Fisher Scientific, Waltham, MA, USA) following manufacturer’s instructions. Eluted peptides from each clean-up were evaporated to dryness in a SpeedVac (Thermo Fisher Scientific, Waltham, MA, USA) and reconstituted in 20 µL 0.1% (*v*/*v*) formic acid. Proteolytic peptide samples were separated by nanoLC using an Ultimate nanoRSLC UPLC and autosampler system (Dionex, Sunnyvale, CA, USA) and analysed on a Tribrid Fusion Lumos mass spectrometer (Thermo Fisher Scientific, Waltham, MA, USA) as described before [27].

### 2.4. Sequence Database Searches and Protein Quantification

Raw peak lists derived from the above experiments were analysed using MaxQuant (version 1.6.2.10) [28] with the Andromeda algorithm [29]. Search parameters were: ±4.5 ppm tolerance for precursor ions and ±0.5 Da for peptide fragments; carbamidomethyl (C) as a fixed modification; oxidation (M) and N-terminal protein acetylation as variable modifications; and enzyme specificity as trypsin with two missed cleavages possible. Peaks were searched against the human Swiss-Prot database (August 2018 release). Label-free protein quantification was performed using the MaxLFQ algorithm with default parameters [30]. Protein and peptide false discovery rate thresholds were set at 1%. Only proteins identified from ≥2 unique peptides and absent from the MaxQuant contaminants database were subjected to downstream analysis. Differential protein abundance analysis was performed in Perseus (version 1.6.5.0) [31]. For each replicate ALI differentiated HBEC culture, protein fold-changes across culture conditions (i.e., across pre-incubation and treatment conditions) were determined for proteins which had been assigned LFQ values in at least two replicates within each condition (missing data imputation was not performed). For each comparison of conditions, *p*-values associated with average log2 protein fold changes were calculated using one sample *t*-tests.

Functional analysis of differentially abundant proteins was performed with IPA (QIAGEN Inc., Germantown, MD, USA) [32]. Mass spectrometry data are available at the ProteomeXchange Consortium via the PRIDE partner repository with the dataset identifier PXD019084 (https://www.ebi.ac.uk/pride/archive/projects/PXD019084). The full list of identified proteins and differentially abundant protein analysis is available upon request.

### 2.5. Measurement of Intracellular Glutathione Levels

Total intracellular glutathione content (reduced plus oxidised glutathione) was measured using a colorimetric glutathione assay kit (Sigma-Aldrich, St. Louis, MO, USA), measured by absorbance at 412 nm using the Versamax Microplate Reader (Molecular Devices, San Jose, CA, USA).

### 2.6. Oxidative Stress Assay

CellROX Green fluorogenic probe (Life Technologies, Carlsbad, CA, USA) was used (according to manufacturer’s instructions) as a surrogate measurement of oxidative stress levels as it is known to detect the ROS superoxide (O_2_^−^) and/or hydroxyl radical (•OH) [33]. The cell-permeable reagents are non- or very weakly-fluorescent while in a reduced state and upon oxidation exhibit strong fluorogenic signal. CellROX^®^ Green Reagent (Thermo Fisher Scientific, Waltham, MA, USA) is a DNA dye, and upon oxidation, it binds to DNA; thus, its signal is localized primarily in the nucleus and mitochondria. The fluorescence intensity (485 nm excitation/520 nm emission) was measured with a EnSight multimode plate reader (Perkin Elmer, Waltham, MA, USA).

### 2.7. Enzyme-Linked Immunosorbent Assay (ELISA)

Cytokine IL-8 content in the cell-free culture media were assessed using an IL-8 ELISA kit (R&D Systems, Minneapolis, MN, USA) as per manufacturer instruction. Samples were diluted 1/10 and all values fell within the standard curve. The ELISA plate was read using the Versamax Microplate Reader (Molecular Devices, San Jose, CA, USA).

### 2.8. High-Performance Liquid Chromatography (HPLC)

IDO-1 dioxygenase activity was determined by measuring the extent of conversion of L-tryptophan (L-Trp) into kynurenine (Kyn) in the culture medium using an Agilent-1260 HPLC system. For this, the cell-free culture medium was treated with 20% trichloroacetic acid in a 3:1 (*v*/*v*) ratio to precipitate proteins and centrifuged (10 min, 18,000× *g*, 4 °C). Protein free supernatants were then injected into a Hypersil 3-µm ODS-C18 column (Phenomenex, Torrance, CA, USA) and eluted at 0.5 mL/min with a mobile phase consisting of 9% (*v*/*v*) acetonitrile and 100 mM chloroacetic acid (pH 2.4). L-Trp and Kyn were detected at 280 and 364 nm, respectively, and their concentrations determined by peak-area comparison of L-Trp and Kyn peaks in experimental samples with authentic L-Trp and Kyn standards of known concentration. IDO-1 activity was presented as the Kyn:L-Trp ratio.

### 2.9. Immunofluorescence and Imaging

Cells were fixed in 3.7% formaldehyde (Thermo Fisher Scientific, Waltham, MA, USA) followed by neutralisation (100 mM glycine for 10 min at room temperature) and permeabilization (0.5% Triton-X for 10 min. The cells were blocked with 10% goat serum (Sigma-Aldrich, St. Louis, MO, USA) for 90 min at room temperature prior to overnight incubation with primary antibodies (Appendix A). Incubation with fluorescently-conjugated secondary antibodies was carried out for 1 h. Cells were mounted with Vectashield hardset antifade medium with DAPI (Vector Laboratories, Burlingame, CA, USA). Tight junctions were immunostained using methanol-acetone fixation for 15 min at −20 °C.

Imaging to validate the cell models was performed using a Leica TCS SP8 DLS confocal microscope with a 20X/0.75 HC PL APO CS2 air objective lens (Leica Microsystems, Wetzlar, Hesse 35576, Germany). Stress granule images were acquired via a Zeiss LSM 780 inverted laser scanning confocal microscope using a 20X/0.8 Plan-Apochromat air objective lens. Quantification of the percentage of cells with stress granules was measured by manually counting the number of cells with ≥3 visible foci per cell. To characterize ZO-1 expression, z-stacks of multiple fields of view (47–55 per treatment) were acquired using Leica TCS SP8 DLS confocal microscope, 63x/1.4 HC PL APO CS2 oil immersion objective (Leica Microsystems, Wetzlar, Hesse 35576, Germany). Images were analysed using custom-built script in Matlab (MathWorks, Natick, MA 01760, USA).

### 2.10. Cell Metabolic Viability Assay (MTT Assay)

Cells were incubated for 24 h with one of the three thiols (GSH, NAC or GGC) with or without LPS at 5% CO_2_ at 37 °C prior to the addition of MTT and 200 µL DMSO to solubilise the cells. Thiazolyl blue tetrazolium bromide (MTT) (Sigma-Aldrich, St. Louis, MO, USA) was added to a final concentration of 0.5 mg/mL. The absorbance of each well was measured at 570 nm using the Versamax Microplate Reader (Molecular Devices, San Jose, CA 95134, USA).

### 2.11. Antioxidant Compounds

GGC sodium salt (Biospecialties International, Mayfield, NSW, 2304., Australia), NAC (Sigma-Aldrich, St. Louis, MO, USA), and GSH (Sigma-Aldrich, St. Louis, MO, USA) were prepared fresh prior to each experiment in cold PBS (Sigma-Aldrich, St. Louis, MO, USA).

### 2.12. Statistical Analysis

The data are presented as a mean with errors representing standard deviation (SD). Single group statistical analysis was performed using one-way ANOVA, while the two-group analysis used two-way ANOVA. All statistical calculations were performed using GraphPad Prism, with the exception of the aforementioned analyses of quantitative proteomics data. *p*-values < 0.05 were considered statistically significant with higher degrees of significance described in figure legends.

## 3. Results

We established primary human airway (bronchial and nasal) epithelial cell models under 2D-conventional and 3D-Air Liquid Interface (ALI) conditions (Figure 1A–C). Immunostaining of 2D-conventional cultures showed formation of confluent monolayers with typical epithelial mosaic-like or cobblestone appearance with cell to cell adhesion bound with adherens junctions and tight-junctions and no detectable signal for mesenchymal cell marker vimentin (Figure 1B). Immunostaining of cells cultured under ALI conditions showed pseudostratified ciliated epithelium, which indicated that the airway epithelial cells were well-differentiated (Figure 1C). MTT assay of CF and non-CF cell metabolic viability showed that treatment of cells with GGC, NAC, and GSH from 2 to 2000 µM did not decrease cell metabolic viability, suggesting their low cytotoxicity (Figure 1D and Appendix A). Treatment with GGC between 2 to 50 µM significantly increased cell metabolic activity, while NAC and GSH treatment did not in both CF and non-CF cultures (Figure 1D and Appendix A). Evaluation of the effect of antioxidant treatment at 50 µM on altering total intracellular glutathione content indicated a significant increase in total intracellular glutathione levels with GGC treatment (Figure 1E). In contrast, incubation with the same concentration of NAC or GSH did not increase total intracellular glutathione levels.

To validate if our model can physiologically recapitulate the oxidative injury on the epithelial cells and the inflammatory microenvironment of the CF respiratory tissue, cells were challenged with *Pseudomonas*-derived LPS. Consistent with previous findings, LPS caused significant induction of oxidative stress [34] and pro-inflammatory cytokine IL-8 production [35] while decreasing cell metabolic viability [36] (Figure 1F–H). We set out to test the mechanism associated with the GGC effect on LPS-challenged CF airway epithelium cultures under six experimental conditions: (1) PBS control (Mock), (2) LPS-challenged (LPS^+^), (3) GGC added 24 h post-LPS challenge (therapeutic, T), (4) GGC added 24 h prior to LPS challenge (prophylactic, P), (5) GGC added 24 h prior and 24 h post-LPS challenge (T+P); and (6) GGC alone (Figure 1I). Parameters including global proteomics alterations, cell redox state (antioxidant (glutathione), oxidative stress), pro-inflammatory mediators (IL-8, IDO-1), and cellular health (membrane integrity, stress granule formation, cell metabolic viability) were assessed (Figure 1J).

### 3.1. LPS Challenge or GGC Treatment Alters Protein Expression in Human Differentiated Airway Epithelium

We first investigated whether the differentiated bronchial epithelium proteome is altered in response to LPS challenge and GGC treatment. Between 1714 and 1944 proteins were identified with quantitative LC-MS/MS in each of the six treatment conditions (Appendix A). To identify significant differentially abundant proteins, pairwise comparisons of conditions was performed. The three GGC treatments (therapeutic, prophylactic, T+P) were compared against the LPS-challenged proteome. LPS-challenged and GGC only proteomes were compared to that of the mock control (Figure 2, Appendix A). In the LPS-challenged comparison, the tight junction protein ZO-1 and immune regulatory and inflammatory proteins (IDO-1, S100A2, CXCL6, and MUC5B) were amongst the most differentially abundant (Figure 2A). In both the therapeutic (Figure 2B) and prophylactic (Figure 2C) comparisons, NAD(P)H-dependant oxidoreductase proteins (BDH2, NDUS3, NDUA8, and AL4A1), the pro-survival protein TCTP, antioxidants peroxiredoxin 3 (PRDX3), and ROS scavenger SERPINB5 were significantly more abundant. Glutathione peroxidase 4 (GPx4), which uses GSH to protect cells against lipid peroxidation, was significantly more abundant in the therapeutic comparison (Figure 2B) condition. In the T+P comparison, three oxidative stress-responsive proteins (CIRBP, PARP1, and APEX1) were differentially abundant (Figure 2D). The antioxidant enzyme, glutaredoxin (GLRX), which employs GSH as a cofactor, was the most significantly altered protein in the comparison of cells treated with GGC alone (Figure 2E).

### 3.2. Biological Pathways Involved in Cellular Respiration, Stress Response and Cell Junction Signalling Are Altered Following the GGC-Treatment of LPS-Challenged Cells

Ingenuity pathway analysis (IPA) [32] of canonical pathways, networks, and biological functions of differentially abundant proteins in each comparison pair was examined (Figure 2F and Appendix A). Altered canonical pathways in the LPS-challenged comparison include oxidative and inflammatory responses such as “NRF2-mediated oxidative stress signalling” and the “IL-17A cytokine signalling pathway”. In both therapeutic and prophylactic treatment comparisons, enriched canonical pathways were related to cellular respiration, protein synthesis, and stress adaptation responses. These same pathways were enriched in the cells treated with GGC alone. Enriched pathways in the T+P condition were involved in the prevention of apoptosis, activation of transcription, and stress response (Figure 2F).

Disease and biological function analysis predicted decreased cell viability in LPS-challenged cells. In contrast, cell viability was predicted to increase in the T+P treatment. Apoptosis and necrosis were predicted to decrease in the prophylactic treatment (Figure 2G).

### 3.3. GGC Reduces Cellular Oxidative Stress but Does Not Alter the Inflammatory Responses in LPS-Challenged Human Differentiated Airway Epithelium

Proteomic analysis revealed an increased abundance of several antioxidant enzymes in differentiated bronchial epithelium with GGC treatment (Figure 2 and Appendix A). We therefore assessed total intracellular glutathione levels in the cells from the six different treatment conditions (Figure 3A). LPS treatment significantly lowered total intracellular glutathione levels by 50% compared to mock control cells. LPS-induced total glutathione depletion was ameliorated by all GGC treatment conditions. Total intracellular glutathione was restored to near unchallenged basal levels. The LPS-induced reduction in total glutathione levels coincided with a marked increase in oxidative stress as detected by oxidation of CellROX Green (Figure 3B). This LPS-induced increase in oxidative stress was significantly decreased with therapeutic, prophylactic, and T+P GGC treatments. (Figure 3B).

IDO-1, a L-tryptophan catabolizing immune regulatory enzyme that is induced during inflammation [37], was significantly upregulated in the proteome of the LPS-challenged cells (Figure 2A). Additionally, the majority of differentially abundant proteins were involved in “antimicrobial response”, and “inflammatory response” networks (Appendix A). In comparison, the three GGC treated conditions did not reveal enrichment in these networks (Appendix A). IDO-1 and pro-inflammatory cytokine IL-8 were assessed to determine whether GGC can alleviate LPS-induced inflammation. LPS challenge resulted in a four-fold increase in IL-8 levels (Figure 3C). A 24-fold increase in cellular IDO-1 enzyme activity (indexed as the kynurenine/tryptophan ratio) was also observed in LPS-treated cells (Figure 3D). However, treatment with GGC did not alter the LPS-induced increases in either IL-8 levels (Figure 3C) or IDO-1 activity (Figure 3D).

### 3.4. GGC Alleviated LPS-Induced Deterioration of Epithelial Tight Junction Protein Zona Occludin-1 (ZO-1) in Human Differentiated Airway Epithelium

Proteomic analysis indicated a significant decrease in bronchial epithelial tight junction protein ZO-1 expression in the LPS-challenged cells (Figure 2A). Furthermore, tight junction and gap junction signalling pathways were enriched in LPS-challenged conditions compared to mock (Figure 2F and Figure 4A). We confirmed this observation by immunofluorescence staining for ZO-1 expression in differentiated airway epithelium (Figure 4B). LPS-challenge significantly reduced the protein expression of ZO-1 in the epithelial cells compared to the unchallenged control cells (Figure 4D). All three GGC treatments ameliorated the LPS-induced reduction in ZO-1 expression. ZO-1 levels were increased above the basal level in therapeutic and T+P. Treatment with GGC in the absence of LPS did not alter ZO-1 expression (Figure 4D).

### 3.5. GGC Prevents LPS-Induced Stress Granule Formation in Human Airway Epithelium

Protein synthesis pathways involved in stress response adaptation were enriched in the differentiated airway epitheliums treated with GGC (Figure 2F and Figure 5A). Differentially abundant proteins in these pathways included G3BP, EIF2S3, EIF3G, and EIF4A1 (Figure 2). These proteins are well-characterised markers of cytosolic stress granules (SG). These granules form when translation initiation is stalled [38]. With the limitation of primary bronchial epithelial proliferation capacity, and since nasal epithelial cultures have been established as an appropriate surrogate for studying lower airway inflammation [39,40] assessment of stress granule formation was carried out in nasal epithelial cultures (HNEC) generated from the same four CF donors. To confirm whether SG formation is modulated by GGC, we assessed the co-localisation of two SG protein markers, G3BP and EIF4A, in cells (Figure 5B). G3BP and EIF4A were seen in a small fraction of cells in all conditions, but they only co-localised in the LPS-challenged cells at low frequency (22% ± 8% cells). Therapeutic treatment of GGC completely ameliorated SG formation. Prophylactic and T+P treatment also prevented LPS-induced SG formation.

### 3.6. Treatment with GGC Improves Metabolic Viability in LPS-Challenged Human Airway Epithelium

Our proteomic analysis revealed modulation of mitochondrial proteins including NADPH-dependent oxidoreductases (Figure 2A), which are enzymes that play a central role in cellular metabolism and catalyse various redox reactions, after treatment with GGC [41]. Assessment of NADPH-dependent oxidoreductase activity is commonly carried out with artificial dyes (Formazan) to determine the metabolic activity of cells which can indicate mitochondrial function and overall cell viability. We assessed airway epithelial cell cultures from CF donors treated with GGC with therapeutic and T+P treatment conditions after the LPS challenge (Figure 6A,B). Treatment with GGC between 2 to 50 µM significantly increased metabolic activity in both treatment conditions.

## 4. Discussion

Developing donor-derived primary airway epithelial differentiated with beating cilia and intact tight junction formation provides a promising in vitro lung model for disease modelling and for lung drug discovery. In this study, we established that the administration of exogenous γ-glutamylcysteine (GGC) to CF airway epithelium in vitro can increase total intracellular glutathione levels and protect cells from LPS-induced cellular damage. Proteomic analysis identified perturbation of several pathways related to cellular respiration, transcription, stress responses, and cell-cell junction signaling upon LPS challenge which were altered when cells were treated with GGC. We confirmed GGC improves cell metabolic viability, tight junction activity, and attenuates LPS-induced oxidative stress and stress granule formation. Our data suggest that GGC therapy could mitigate the lung tissue damage suffered by CF patients from repeated *P. aeruginosa* infections.

CF is characterised by persistent inflammation, oxidative stress of the airways, and deficiency in the extracellular antioxidant glutathione [42]. Despite multiple clinical trials with current glutathione-based therapies, improved clinical outcomes in CF have not been achieved [6,7]. As a precursor to glutathione, GGC has been consistently shown to increase intracellular levels of glutathione in both in vivo and in vitro models [16,17,18,21,22,23,24], and to reduce inflammation and oxidative stress [19,43,44]. Recently, it was shown that GGC exhibits anti-inflammatory effects in an in vivo and in vitro mouse sepsis model [24]. However, in our airway ALI CF model, administration of GGC had no impact on reducing the LPS-challenged upregulation of two inflammatory markers, IL-8 and IDO-1. While inflammation and oxidative stress are tightly linked processes that can induce one another, they can also occur independently [35,45,46].

GGC administration significantly increased total intracellular glutathione levels in our LPS-challenged airway CF model. Both prophylactic and therapeutic treatment with GGC were effective, with a combination of both being the most effective measure. This concurred with a previous study that showed GGC supplementation ameliorates glutathione depletion caused by LPS in a sepsis mouse model [24].

GGC itself is an antioxidant by acting as an alternative co-factor for glutathione peroxidase (GPx), regardless of changes to the intracellular GSH concentration [19]. We found elevated GPx4 expression in LPS-challenged cells that had been treated with GGC. GPx4 is an essential antioxidant protein that protects against oxidative stress by degrading the ROS hydrogen peroxide and lipid hydroperoxides [47,48]. Peroxidation of membrane lipids can disturb the assembly of cell membranes, which inevitably will impact membrane permeability. Intestinal malabsorption of fat-soluble vitamins, a characteristic of CF, causes a deficiency of the major lipid membrane antioxidant vitamin E leading to excessive lipid peroxidation [5]. Further impairment of the membrane integrity is evident by dislocation of epithelial tight junction protein ZO-1 from the plasma membrane to the cytosol and nucleus in CF epithelial cells [49]. ZO-1 expression is further reduced with LPS challenge [50]. Downregulation of ZO-1 in LPS-challenged CF cells was evident in both our proteomics and immunofluorescence analyses. GGC treatment post-LPS challenge showed significant reestablishment of ZO-1 expression at the cell boundaries. Pre-treatment with GGC protected against LPS-induced ZO-1 reduction, though not to the same magnitude as that observed for the therapeutic or combined therapeutic and prophylactic GGC treatment. This suggested that treatment with GGC in either prophylactic or therapeutic capacity may be effective for reversing LPS-induced and persistent CF-related membrane integrity impairment, protecting against viral or bacterial infection.

Inhibition of protein synthesis is a well-characterised consequence of exposure of cells to oxidative stress [38]. We observed alterations in protein synthesis pathways, including the EIF2 signalling pathway, in the proteomes of CF differentiated airway epithelium. Inhibition of translation initiation leads to the accumulation of stalled translation pre-initiation complexes (PIC) that condense to form non–membrane-enclosed foci known as stress granules (SG). SG formation can be triggered by protein misfolding. No SGs were observed in the mock or control cells, which suggested that misfolded mutant CFTR does not cause chronic SG formation. LPS challenge is known to stimulate SG formation [51]. In our model, exposure to LPS caused SG formation in approximately 20% of cells. Elevated levels of GSH has been shown to inhibit arsenite-induced SG formation in West Nile virus-infected cells [52]. Our data demonstrated that GGC effectively inhibits LPS-induced SG formation in CF cells in the therapeutic, prophylactic, and combination treatment regimens. The inhibition of SG formation supports that elevation of total intracellular glutathione levels by GGC treatment protects against LPS-induced inhibition of translation initiation, thereby preserving cellular protein synthesis.

We observed protection against LPS-induced metabolic dysfunction in all GGC treatments in CF airway epithelial cells, suggesting an overall improvement in cell metabolic viability. Even in the absence of LPS, treatment with GGC improved metabolic activity. GGC may alleviate mitochondrial metabolic dysfunction caused by defective CFTR, which is known to induce excessive ROS production and promote *P. aeruginosa* infection [53]. In contrast, NAC and GSH supplementation did not demonstrate significant improvements in unchallenged cells, potentially explaining the lack of success for GSH and NAC clinical trials in improving pulmonary function [7,8]. Increased metabolic activity may in part be due to increased expression of NAD(P)H-dependant oxidoreductase proteins observed in the proteome after both therapeutic and prophylactic treatment with GGC. We note that treatment at the exceptionally high concentration of 2000 μM with GGC compromised metabolic viability in the therapeutic and prophylactic combined condition. This concentration equates to 17.5 g of an in vivo dose. Animal safety trials have demonstrated that GGC is safe at a repeated daily dose at a limit dosage of 1000 mg/kg over a 90-day period [20]. A single dose of oral administered GGC at 2 g and 4 g is bioavailable and can increase intracellular GSH levels above homeostasis in lymphocytes with no adverse effects [16]. Therefore, this concentration may represent an abnormally high dosage of GGC that provides no therapeutic value.

Our data showed that GGC treatment of CF human airway cells increases the overall cellular redox status of the cell in favour of a less oxidative state, which may alleviate LPS-induced cell oxidative stress and ROS production. This may occur through multiple interplaying mechanisms: (1) increasing total intracellular glutathione levels which directly scavenge ROS; (2) acting as a reducing co-factor for antioxidant proteins (e.g., GPx4); and (3) upregulating oxidative stress response proteins (e.g., peroxiredoxins). We provide promising data in support of a beneficial effect of GGC for CF antioxidant therapy. GGC has self-affirmed Generally Recognised as Safe (GRAS) status, which should facilitate and simplify its regulatory pathway to the clinic. Products containing GGC for oral consumption are now on sale to the general public in the USA. An oral route of delivery allows for rapid first-pass metabolism of GGC via the gut and liver and a subsequent increase in circulating GGC [16]. Defects in gut and liver function in CF patients, however, could hamper oral GGC efficacy. Controlled clinical studies are now needed to investigate GGC’s safety in CF patients and its potential as a useful preventative and therapeutic treatment for CF airway redox imbalance.

## Figures and Tables

**Figure 1 antioxidants-09-01204-f001:**
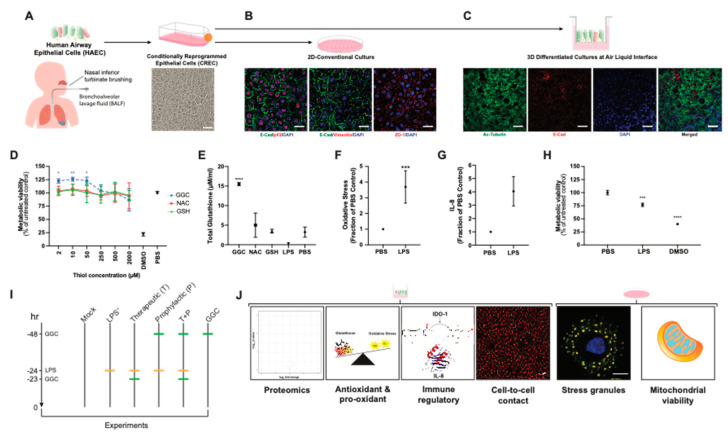
Validation of physiological response of Cystic Fibrosis (CF) derived differentiated airway epithelium model to LPS challenge. (**A**) Airway basal epithelial cells from four CF donors were used to create conditionally reprogrammed airway epithelial cells (CREC). (**B**) CRECs were cultured under 2D-conventional condition which formed epithelial monolayers that retain their epithelial cobblestone morphology (positive E-Cadherin (green), ZO-1 (red), and p63 (red) staining) and no detectable signal for mesenchymal cell marker vimentin (VIM; red). (**C**) Alternatively, CRECs were cultured in 3D terminally differentiated pseudostratified epithelium at air liquid interface (ALI) with mucociliary differentiation (ciliated cell marker Ac-Tubulin (green) and E-cadherin (red)). (**D**) Treatment of epithelial cultures with GGC, NAC, and GSH for 24 h in a range of concentration (2 to 2000 µM) shows low cytotoxicity. DMSO was used as positive cell death control. Cytotoxicity was observed as a reduction in metabolic viability measured by MTT assay. (**E**) Total intracellular glutathione levels increase with 50 µM GGC but not 50 µM of NAC or GSH. Exposure of epithelial cultures to 100 ug/mL of *Pseudomonas aeruginosa* derived lipopolysaccharide (LPS) for 24 h show significant increase in (**F**) oxidative stress and (**G**) inflammation via a four-fold increase in pro-inflammatory cytokine IL-8 and (**H**) increased cytotoxicity. (**I**) Treatments used to explore the potential efficacy of GGC against *Pseudomonas aeruginosa* derived lipopolysaccharide (LPS) included: (1) PBS control (Mock), (2) LPS-challenged (LPS+), (3) GGC added 24 h post-LPS challenge (therapeutic, T), (4) GGC added 24 h prior to LPS challenge (prophylactic, P), (5) GGC added 24 h prior and 24 h post LPS challenge (T+P); and (6) GGC alone. (**J**) Proteomics, antioxidant, and pro-oxidant, inflammatory response, cell morphology and metabolic viability analyses were carried out 24 h post-infection for each treatment condition in primary differentiated human airway epithelial cells (HAEC) of either bronchial (HBEC) or nasal (HNEC) origin. Statistical analysis was performed using one-way ANOVA against the PBS control. (* = *p* < 0.05, ** = *p* < 0.01, *** = *p* < 0.001, **** = *p* < 0.0001). Error bars represent SD. Data are from at least three primary airway CF (DF508/DF508 CFTR) epithelium cultures, with three technical replicates for each different experimental condition.

**Figure 2 antioxidants-09-01204-f002:**
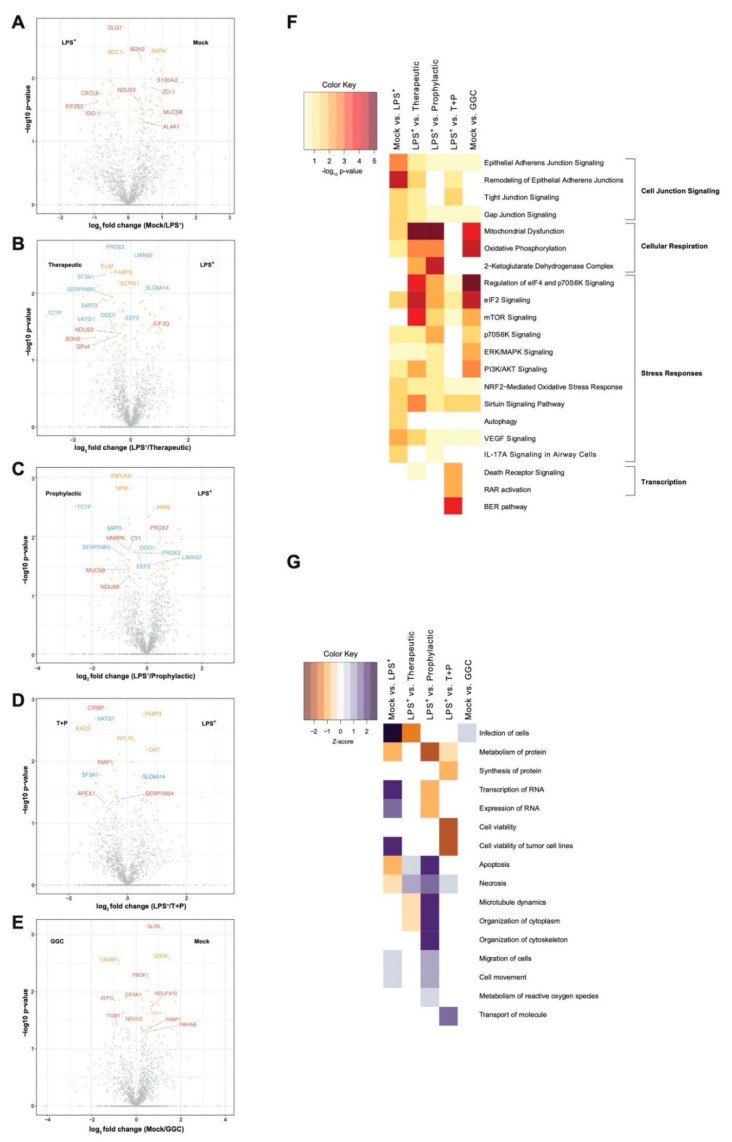
Differential protein abundance in differentiated human bronchial epithelial cells after LPS challenge and GGC treatment. Proteomes of four cystic fibrosis (CF) derived differentiated airway epithelium cultures with three technical replicates were explored for the potential efficacy of GGC against LPS challenge for (1) PBS control (Mock), (2) LPS-challenged (LPS^+^), (3) GGC added 24 h post-LPS challenge (therapeutic, T), (4) GGC added 24 h prior to LPS challenge (prophylactic, P), (5) GGC added 24 h prior and 24 h post LPS challenge (T+P); and (6) GGC alone experimental conditions. Volcano plots of differentially abundant proteins between pairwise comparisons of: (**A**) mock and LPS^+^, (**B**) LPS^+^ and therapeutic treatment, (**C**) LPS^+^ and prophylactic treatment, (**D**) LPS^+^ and T+P treatment, and (**E**) mock and GGC only conditions. The log_10_ transformed *p*-values (*y*-axis) were plotted against the average log_2_ fold-change (*x*-axis). Proteins represented by orange dots were statistically significant (*p* < 0.05) and grey dots were not statistically significant. Proteins of biological interest (red) and differentially abundant in multiple comparisons (blue) are labelled. (**F**) A heatmap of the enriched canonical pathways among differentially abundant proteins in the five comparisons (**A**–**E**) discovered by Ingenuity Pathway Analysis (IPA). Colour indicates-log_10_
*p*-value of pathways with high statistical significance (red) and low statistical significance (yellow). (**G**) Heatmap of enriched disease biological functions among differentially abundant proteins in the five comparisons (**A**–**E**) discovered by IPA. Colour indicates the activation Z-score and predicts whether a disease or biological function is increased (positive Z-score; blue) or decreased (negative Z-score; brown) based on the experimental dataset. Data are from four CF derived differentiated airway epithelium cultures with three technical replicates for each different experimental condition.

**Figure 3 antioxidants-09-01204-f003:**
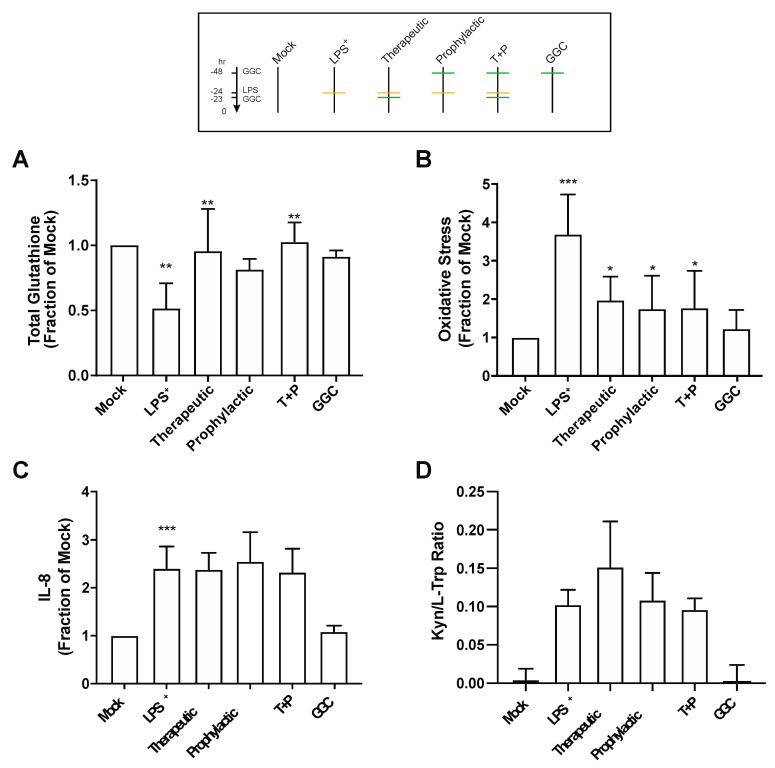
GGC ameliorates LPS-induced reduction in intracellular glutathione levels and increase in cellular oxidative stress levels, but not LPS-induced inflammatory and immune regulatory responses. Cells from mock, LPS^+^, therapeutic (T), prophylactic (P), pre and post LPS^+^ treated (T+P), and GGC only conditions were assayed for (**A**) total intracellular glutathione, (**B**) oxidative stress levels, (**C**) IL-8 cytokine, and (**D**) IDO-1 activity measured as the kynurenine to tryptophan ratio. Statistical analysis was performed using two-way ANOVA against LPS^+^ for therapeutic, prophylactic and T+P conditions, and mock for LPS^+^ and GGC only (* = *p* < 0.05, ** = *p* < 0.01, *** = *p* < 0.001). Error bars represent SD. Data are from four cystic fibrosis (CF) derived differentiated airway epithelium cultures with three technical replicates for each different experimental condition.

**Figure 4 antioxidants-09-01204-f004:**
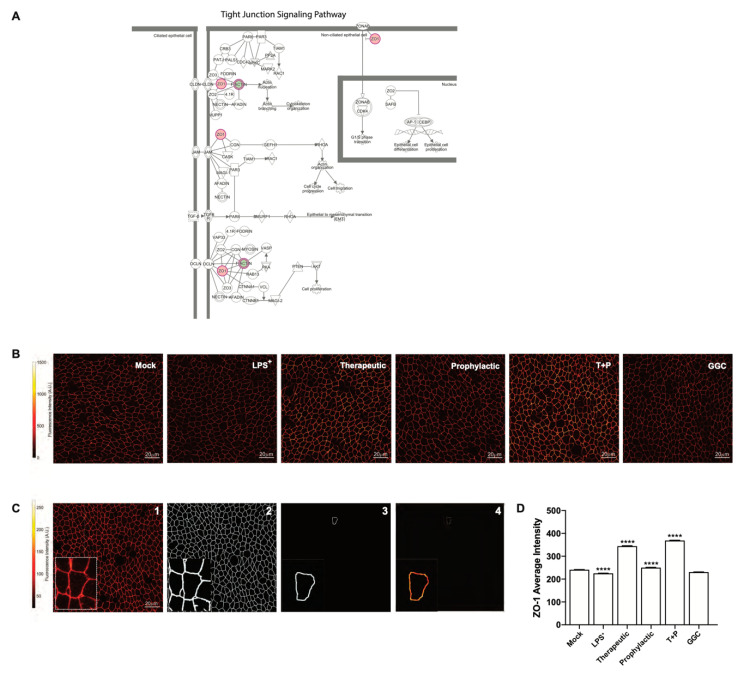
Reduction of tight junction protein ZO-1 by LPS challenge was ameliorated with GGC. Mock, LPS^+^, therapeutic (T), prophylactic (P), pre and post LPS^+^ treated (T+P), and GGC only conditions were assessed for tight junction protein ZO-1 expression. (**A**) Tight junction signalling pathway. Red highlighted proteins were more abundant in LPS^+^ relative to the mock, while green was less abundant in LPS^+^ relative to the mock. The pathway was adapted from IPA (QIAGEN). (**B**) Representative image of ZO-1 immunofluorescence in the six experimental conditions. (**C**) Summary workflow of ZO-1 quantification. The mid-plane of acquired z-stack images (plane with the highest intensity) were determined (panel 1) and used to generate the binary mask of tight junctions (panel 2). The junction of individual cells was extracted (panel 3) and the mid-plane intensity of pixels spanning the entire cellular junction was used to calculate the average intensity and standard deviation of intensity within each cell (panel 4). (**D**) Mean ZO-1 intensity as calculated using the above workflow. Data are from four cystic fibrosis (CF) derived differentiated airway epithelium cultures. Statistical analysis was performed using one-way ANOVA against LPS^+^ for T, P, and T+P conditions, and mock for LPS^+^ and GGC only (**** = *p* < 0.0001). Error bars represent SD. Scale bars represent 20 µm.

**Figure 5 antioxidants-09-01204-f005:**
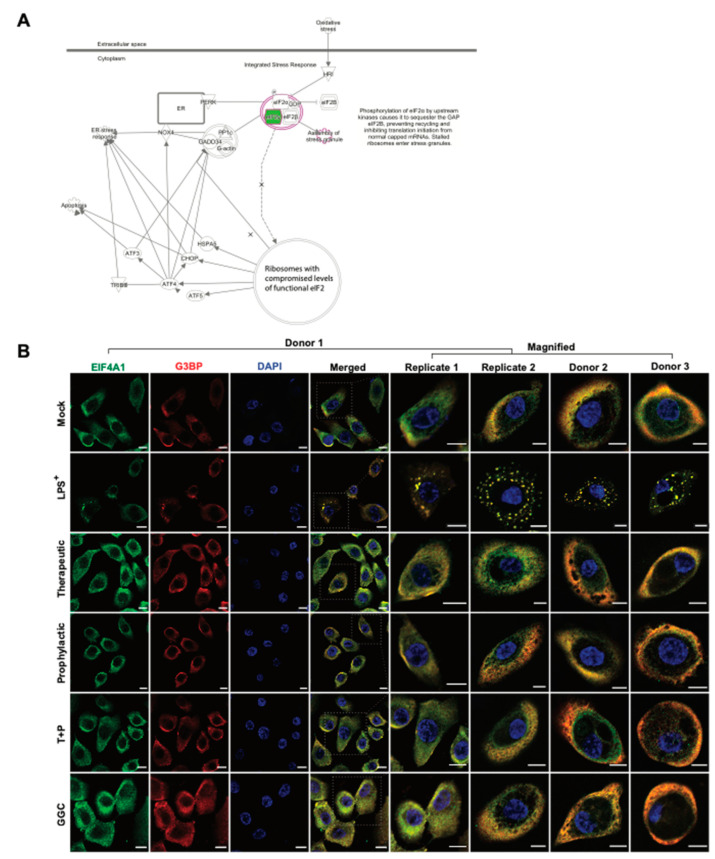
GGC inhibited LPS-induced stress granule formation. Mock, LPS^+^, therapeutic (T), prophylactic (P), pre and post LPS^+^ treated (T+P), and GGC only conditions were assessed stress granule protein localisation. (**A**) The EIF2 signalling pathway, which is involved in stress response adaptation including the assembly of stress granules. Green highlighted proteins were less abundant in LPS^+^ relative to the mock (Figure 3A). The pathway is adapted from IPA (QIAGEN). (**B**) Representative immunofluorescence co-staining for stress granule markers EIF4A and G3BP images from three cystic fibrosis (CF) donors’ primary nasal epithelial cultures (HNEC) in the six experimental conditions. Additionally, HNEC cells from Donor 1 were cultured on two separate occasions and assessed independently (Replicate 1 and 2). Stress granules were identified in cells where both EIF4A (green) and G3BP (red) signals overlapped. Nuclei are stained blue. Scale bars represent 10 µm.

**Figure 6 antioxidants-09-01204-f006:**
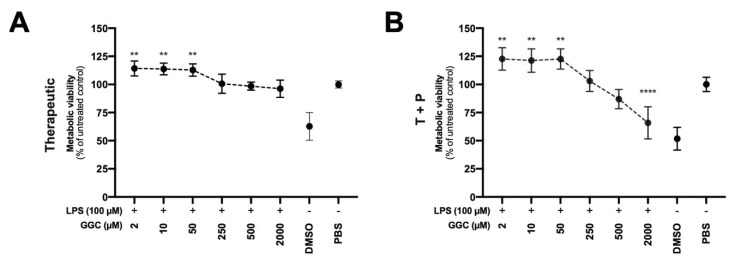
GGC increases the metabolic viability of LPS-challenged human airway epithelial cells. Cell metabolic viability was assessed in (**A**) post-LPS^+^ treated (therapeutic) and (**B**) pre and post LPS^+^ treated (T+P) conditions in cystic fibrosis (CF) human airway epithelial cells (HAEC) using the MTT assay. GGC treatment in a range of concentration (2 to 2000 µL) were tested. DMSO used as positive cell death control. Statistical analysis was performed using one-way ANOVA against LPS challenged cells. (** = *p* < 0.01, **** = *p* < 0.0001). Error bars represent SD. Data are from independent experiments performed on CF airway epithelium cultures from three primary human nasal epithelial cells and two immortalised human bronchial epithelial cells, with three technical replicates for each different experimental condition.

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
