# Peer review of "Novel Antioxidant Therapy with the Immediate Precursor to Glutathione, γ-Glutamylcysteine (GGC), Ameliorates LPS-Induced Cellular Stress in In Vitro 3D-Differentiated Airway Model from Primary Cystic Fibrosis Human Bronchial Cells"

_antioxidants, 2020, doi:10.3390/antiox9121204_

Round 1

Reviewer 1 Report

Summary

This study tests if γ-glutamylcysteine (GGC) prevents/reverses molecular/cellular phenotypes of LPS-induced injury in airway epithelial cells from patients with cystic fibrosis. Studies focused on GGC since glutathione (GSH) transport can become compromised due to mutations in the cystic fibrosis transmembrane regulator (CFTR). To increase intracellular GSH levels, GGC will be imported efficiently for synthesis. A proteomic approach was used to identify differentially-expressed proteins and molecular/disease pathways during GGC treatment in LPS-exposed cells. Data suggest that GGC alleviated LPS-induced oxidative stress (reduced GSH, increased CellROX signal) and was associated with specific molecular pathways based on differential protein expression which were validated (tight junctions, stress granules). While this work has added value, there are several major gaps of knowledge for this study which require attention/clarification.

Major concerns

  1. The hypothesis is confusing. A prior study (PMID 28982600) is cited as demonstrating that cystic fibrosis is characterized by GSH deficiency. This study shows decreased total glutathione, GSH, and GSSG in bronchoalveolar lavage from children with and without cystic fibrosis which could be explained by impaired GSH efflux due to CFTR mutations. This work doesn’t inform on intracellular GSH levels although the authors hypothesize GGC as a potential therapy to increase intracellular GSH levels. Are there baseline and/or LPS-dependent differences in GSH levels in airway epithelial cells isolated from patients with or without cystic fibrosis?
  2. Your method of GSH detection measures both GSH and GSSG (total glutathione). However, redox potential is best represented quantifying reduced and oxidized components of the redox couple (GSH:GSSG). Were GSH and GSSG measured? If not, presentation of the data is misleading because it is total glutathione and not just reduced glutathione.
  3. What bioinformatics workflows were used to determine changes between conditions? For example, how were contaminants removed (e.g., CRAPome); were peptides only considered if they were detected across all four cell lines; what analyses were performed to constitute significance?
  4. Concerned that “NaN” appears for numerous samples in Table S3 and assuming that this means the peptide wasn’t detected during one of the comparative conditions. Therefore, you only have two or three biological replicates for many of your peptides instead of all four cell lines.
  5. Why weren’t control cells treated with LPS to determine DF508-CFTR-dependent effects. For example, section 3.4 highlights LPS-dependent changes in ZO-1 expression but there’s no way to know if this is response is specific for this CFTR mutation or not. Similar concern for stress granule formation.
  6. Why are high levels of GGC toxic in the T+P treatment model? (Fig 6B)

Minor concerns

  1. What were the ages/sex of the patients during brushing for airway epithelial cells? Age/sex could be important variables.
  2. Is LPS-dependent decrease in metabolic activity due to molecular/cellular pathways or cell death? Was cell death quantified?
  3. Page 6, lines 240-1: should mention metabolic activity changes in both CFTR and non-CFTR donor cells.
  4. Page 7, lines 272-3: cells are challenged or treated with LPS (not infected).

Reviewer 2 Report

In this paper, authors deal with an hot topic of cystic fibrosis: Improvements to current therapeutic strategies. Airway oxidative stress and inflammation was induced by P. aeruginosa lipopolysaccharide (LPS). Global proteomics alterations, cell redox state (glutathione, ROS), pro-inflammatory mediators (IL-8, IDO-1) and cellular health (membrane integrity, stress granule formation, cell metabolic viability) were assayed under different experimental conditions revealing alteration of of several pathways related to cellular respiration and stress responses upon LPS challenge.  Both therapeutic and prophylactic treatments were successful, indicate that GGC has therapeutic potential for treatment and prevention of oxidative stress related damage to airways in Cystic Fibrosis.

The manuscript results well written and clear. The use of human cells increase the interest of the paper, as well as the use of 3D cell culture. I think that very the increasing number of paper that speak about the importance of Antioxydants in CF, support the importance of this topis. This results were also in line with journal aim and scope.

Author Response

Response to reviewer 2

Reviewer 2

In this paper, authors deal with an hot topic of cystic fibrosis: Improvements to current therapeutic strategies. Airway oxidative stress and inflammation was induced by P. aeruginosa lipopolysaccharide (LPS). Global proteomics alterations, cell redox state (glutathione, ROS), pro-inflammatory mediators (IL-8, IDO-1) and cellular health (membrane integrity, stress granule formation, cell metabolic viability) were assayed under different experimental conditions revealing alteration of of several pathways related to cellular respiration and stress responses upon LPS challenge.  Both therapeutic and prophylactic treatments were successful, indicate that GGC has therapeutic potential for treatment and prevention of oxidative stress related damage to airways in Cystic Fibrosis.

The manuscript results well written and clear. The use of human cells increase the interest of the paper, as well as the use of 3D cell culture. I think that very the increasing number of paper that speak about the importance of Antioxydants in CF, support the importance of this topis. This results were also in line with journal aim and scope.

Our response: We would like to thank the reviewer for the very positive review. The reviewer did not have any specific comments or concerns. As such, no changes have been made in regard to the comment from reviewer 2.

Reviewer 3 Report

The article by Hewson and co-workers reports a novel observation that surely fosters the investigation of potential therapeutic and/or prophylactic treatments of oxidative stress related damage in CF.

The paper is well-written and contains interesting data acquired with appropriate cell models, therefore it is worth publishing in Antioxidants.

Author Response

Response to reviewer 3

Comments and Suggestions for Authors

The article by Hewson and co-workers reports a novel observation that surely fosters the investigation of potential therapeutic and/or prophylactic treatments of oxidative stress related damage in CF.

The paper is well-written and contains interesting data acquired with appropriate cell models, therefore it is worth publishing in Antioxidants.

Our response: We would like to thank the reviewer for the very positive review. The reviewer did not have any specific comments or concerns. As such, no changes have been made in regard to the comment from reviewer 3.

Round 2

Reviewer 1 Report

Summary

The authors edited their manuscript to consider numerous suggestions based on the prior comments. Although there is still some disconnect between glutathione localization (intracellular/extracellular) in their hypothesis and experimental approach, the authors now focus on testing if γ-glutamylcysteine (GGC) has therapeutic efficacy against LPS-induced injury in airway epithelial cells from patients with cystic fibrosis since intracellular glutathione is typically depleted. Since the glutathione redox couple (GSH:GSSG) is used as an indicator of pathogenesis in numerous diseases, the manuscript now focuses on total glutathione since there was no discrimination between GSH and GSSG. Although quantification of the redox couple provides much greater mechanistic insight, there is still value in evaluating total glutathione during this phenomenon. Overall, the manuscript is much improved after these edits and there are still some clarifications that may require attention.  

Minor concerns

  1. Be very cautious regarding interpretations of ROS and cellular redox status using CellROX – misuse of non-specific redox-sensitive fluorescent dyes simplifies very complex redox chemistries highly regulated by biological systems and/or results in gross misinterpretation of data (see PMID 22027063 for review). First, all reactive/reduced oxygen species are unique and it is inaccurate to cluster/discuss as a single entity. Second, you are not measuring the redox state of the cell, only the redox state of your surrogate detection molecular, CellROX. Lastly, since the oxidative mechanism of CellROX is not known (to my knowledge), it is possible that ROS do not directly oxidize CellROX (much like DCFH-DA) and that other reactive molecular regulate CellROX redox status.
  2. Page 3, line 103: “…GGC can be take up by intact cells…”
